# Stein Variational Gradient Descent as Gradient Flow

**Qiang Liu**
Department of Computer Science
Dartmouth College
Hanover, NH 03755
qiang.liu@dartmouth.edu

## Abstract

Stein variational gradient descent (SVGD) is a deterministic sampling algorithm that iteratively transports a set of particles to approximate given distributions, based on a gradient-based update that guarantees to optimally decrease the KL divergence within a function space. This paper develops the first theoretical analysis on SVGD. We establish that the empirical measures of the SVGD samples weakly converge to the target distribution, and show that the asymptotic behavior of SVGD is characterized by a nonlinear Fokker-Planck equation known as Vlasov equation in physics. We develop a geometric perspective that views SVGD as a gradient flow of the KL divergence functional under a new metric structure on the space of distributions induced by Stein operator.

## 1  Introduction

Stein variational gradient descent (SVGD) [1] is a particle-based algorithm for approximating complex distributions. Unlike typical Monte Carlo algorithms that rely on randomness for approximation, SVGD constructs a set of points (or particles) by iteratively applying *deterministic* updates that is constructed to optimally decrease the KL divergence to the target distribution at each iteration. SVGD has a simple form that efficient leverages the gradient information of the distribution, and can be readily applied to complex models with massive datasets for which typical gradient descent has been found efficient. A nice property of SVGD is that it strictly reduces to the typical gradient ascent for maximum a posteriori (MAP) when using only a single particle ($n = 1$), while turns into a full sampling method with more particles. Because MAP often provides reasonably good results in practice, SVGD is found more *particle-efficient* than typical Monte Carlo methods which require much larger numbers of particles to achieve good results.

SVGD can be viewed as a variational inference algorithm [e.g., 2], but is significantly different from the typical *parametric* variational inference algorithms that use parametric sets to approximate given distributions and have the disadvantage of introducing deterministic biases and (often) requiring non-convex optimization. The non-parametric nature of SVGD allows it to provide consistent estimation for generic distributions like Monte Carlo does. There are also particle algorithms based on optimization, or variational principles, with theoretical guarantees [e.g., 3–5], but they often do not use the gradient information effectively and do not scale well in high dimensions.

However, SVGD is difficult to analyze theoretically because it involves a system of particles that interact with each other in a complex way. In this work, we take an initial step towards analyzing SVGD. We characterize the SVGD dynamics using an evolutionary process of the empirical measures of the particles that is known as Vlasov process in physics, and establish that empirical measures of the particles weakly converge to the given target distribution. We develop a geometric interpretation of SVGD that views SVGD as a gradient flow of KL divergence, defined on a new Riemannian-like metric structure imposed on the space of density functions.

## 2   Stein Variational Gradient Descent (SVGD)

We start with a brief overview of SVGD [1]. Let $\nu_p$ be a probability measure of interest with a positive, (weakly) differentiable density $p(x)$ on an open set $X \subseteq \mathbb{R}^d$. We want to approximate $\nu_p$ with a set of particles $\{x_i\}_{i=1}^n$ whose empirical measure $\hat{\mu}_n(\mathrm{d}x) = \sum_{i=1}^n \delta(x - x_i)/n\,\mathrm{d}x$ weakly converges to $\nu_p$ as $n \to \infty$ (denoted by $\hat{\mu}_n \Rightarrow \nu_p$), in the sense that we have $\mathbb{E}_{\hat{\mu}_n}[h] \to \mathbb{E}_{\nu_p}[h]$ as $n \to \infty$ for all bounded, continuous test functions $h$.

To achieve this, we initialize the particles with some simple distribution $\mu$, and update them via map

$$\boldsymbol{T}(x) = x + \epsilon\boldsymbol{\phi}(x),$$

where $\epsilon$ is a small step size, and $\phi(x)$ is a perturbation direction, or velocity field, which should be chosen to maximally decrease the KL divergence of the particle distribution with the target distribution; this is framed by [1] as solving the following functional optimization,

$$\max_{\phi \in \mathcal{H}} \left\{ -\frac{\mathrm{d}}{\mathrm{d}\epsilon}\mathrm{KL}(\boldsymbol{T}\mu \,||\, \nu_p)\,\big|_{\epsilon=0} \quad s.t. \quad ||\boldsymbol{\phi}||_{\mathcal{H}} \le 1 \right\}. \tag{1}$$

where $\mu$ denotes the (empirical) measure of the current particles, and $\boldsymbol{T}\mu$ is the measure of the updated particles $x' = \boldsymbol{T}(x)$ with $x \sim \mu$, or the pushforward measure of $\mu$ through map $\boldsymbol{T}$, and $\mathcal{H}$ is a normed function space chosen to optimize over.

A key observation is that the objective in (1) is a linear functional of $\phi$ that draws connections to ideas in the Stein's method [6] used for proving limit theorems or probabilistic bounds in theoretical statistics. Liu and Wang [1] showed that

$$-\frac{\mathrm{d}}{\mathrm{d}\epsilon}\mathrm{KL}(\boldsymbol{T}\mu \,||\, \nu_p)\big|_{\epsilon=0} = \mathbb{E}_\mu[\mathcal{S}_p\boldsymbol{\phi}], \quad \text{with} \quad \mathcal{S}_p\boldsymbol{\phi}(x) \coloneqq \nabla \log p(x)^\top \boldsymbol{\phi}(x) + \nabla \cdot \boldsymbol{\phi}(x), \tag{2}$$

where $\nabla \cdot \boldsymbol{\phi} \coloneqq \sum_{k=1}^d \partial_{x_k}\phi_k(x)$, and $\mathcal{S}_p$ is a linear operator that maps a vector-valued function $\phi$ to a scalar-valued function $\mathcal{S}_p\phi$, and $\mathcal{S}_p$ is called the *Stein operator* in connection with the so-called *Stein's identity*, which shows that the RHS of (2) equals zero if $\mu = \nu_p$,

$$\mathbb{E}_p[\mathcal{S}_p\boldsymbol{\phi}] = \mathbb{E}_p[\nabla \log p^\top \boldsymbol{\phi} + \nabla \cdot \boldsymbol{\phi}] = \int \nabla \cdot (p\boldsymbol{\phi})\mathrm{d}x = 0; \tag{3}$$

it is the result of integration by parts, assuming proper zero boundary conditions. Therefore, the optimization (1) reduces to

$$\mathbb{D}(\mu \,||\, \nu_p) \coloneqq \max_{\phi \in \mathcal{H}} \left\{ \mathbb{E}_\mu[\mathcal{S}_p\boldsymbol{\phi}], \quad s.t. \quad ||\boldsymbol{\phi}||_{\mathcal{H}} \le 1 \right\}, \tag{4}$$

where $\mathbb{D}(\mu \,||\, \nu_p)$ is called *Stein discrepancy*, which provides a discrepancy measure between $\mu$ and $\nu_p$, since $\mathbb{D}(\mu \,||\, \nu_p) = 0$ if $\mu = \nu_p$ and $\mathbb{D}(\mu \,||\, \nu_p) > 0$ if $\mu \ne \nu_p$ given $\mathcal{H}$ is sufficiently large.

Because (4) induces an infinite dimensional functional optimization, it is critical to select a nice space $\mathcal{H}$ that is both sufficiently rich and also ensures computational tractability in practice. Kernelized Stein discrepancy (KSD) provides one way to achieve this by taking $\mathcal{H}$ to be a reproducing kernel Hilbert space (RKHS), for which the optimization yields a closed form solution [7–10].

To be specific, let $\mathcal{H}_0$ be a RKHS of scalar-valued functions with a positive definite kernel $k(x, x')$, and $\mathcal{H} = \mathcal{H}_0 \times \cdots \times \mathcal{H}_0$ the corresponding $d \times 1$ vector-valued RKHS. Then it can be shown that the optimal solution of (4) is

$$\boldsymbol{\phi}_{\mu,p}^*(\cdot) \propto \mathbb{E}_{x \sim \mu}[\mathcal{S}_p \otimes k(x, \cdot)], \quad \text{with} \quad \mathcal{S}_p \otimes k(x, \cdot) \coloneqq \nabla \log p(x)k(x, \cdot) + \nabla_x k(x, \cdot), \tag{5}$$

where $\mathcal{S}_p\otimes$ is an outer product variant of Stein operator which maps a scalar-valued function to a vector-valued one. Further, it has been shown in [e.g., 7] that

$$\mathbb{D}(\mu \,||\, \nu_p) = ||\boldsymbol{\phi}_{\mu,p}^*||_{\mathcal{H}} = \sqrt{\mathbb{E}_{x,x' \sim \mu}[\kappa_p(x, x')]}, \quad \text{with} \quad \kappa_p(x, x') \coloneqq \mathcal{S}_p^x \mathcal{S}_p^{x'} \otimes k(x, x'), \tag{6}$$

where $\kappa_p(x, x')$ is a "Steinalized" positive definite kernel obtained by applying Stein operator twice; $\mathcal{S}_p^x$ and $\mathcal{S}_p^{x'}$ are the Stein operators w.r.t. variable $x$ and $x'$, respectively. The key advantage of KSD is its computational tractability: it can be empirically evaluated with samples drawn from $\mu$ and the gradient $\nabla \log p$, which is independent of the normalization constant in $p$ [see 7, 8].

---

**Algorithm 1** Stein Variational Gradient Descent [1]

---

**Input:** The score function $\nabla_x \log p(x)$.
**Goal:** A set of particles $\{x^i\}_{i=1}^n$ that approximates $p(x)$.
**Initialize** a set of particles $\{x_0^i\}_{i=1}^n$; pick a positive definite kernel $k(x, x')$ and step-size $\{\epsilon_\ell\}$.
**For** iteration $\ell$ **do**

$$x_{\ell+1}^i \leftarrow x_\ell^i + \epsilon \phi_{\hat{\mu}_\ell^n, p}^*(x_\ell^i), \quad \forall i = 1, \dots, n,$$

$$\text{where} \quad \phi_{\hat{\mu}_\ell^n, p}^*(x) = \frac{1}{n} \sum_{j=1}^n \left[ \nabla \log p(x_\ell^j) k(x_\ell^j, x) + \nabla_{x_\ell^j} k(x_\ell^j, x) \right], \tag{8}$$

---

An important theoretic issue related to KSD is to characterize when $\mathcal{H}$ is rich enough to ensure $\mathbb{D}(\mu \,\|\, \nu_p) = 0$ iff $\mu = \nu_p$; this has been studied by Liu et al. [7], Chwialkowski et al. [8], Oates et al. [11]. More recently, Gorham and Mackey [10] (Theorem 8) established a stronger result that Stein discrepancy implies weak convergence on $X = \mathbb{R}^d$: let $\{\mu_\ell\}_{\ell=1}^\infty$ be a sequence of probability measures, then

$$\mathbb{D}(\mu_\ell \,\|\, \nu_p) \to 0 \iff \mu_\ell \Rightarrow \nu_p \quad \text{as } \ell \to \infty, \tag{7}$$

for $\nu_p$ that are distantly dissipative (Definition 4 of Gorham and Mackey [10]) and a class of inverse multi-quadric kernels. Since the focus of this work is on SVGD, we will assume (7) holds without further examination.

In SVGD algorithm, we iteratively update a set of particles using the optimal transform just derived, starting from certain initialization. Let $\{x_\ell^i\}_{i=1}^n$ be the particles at the $\ell$-th iteration. In this case, the exact distributions of $\{x_\ell^i\}_{i=1}^n$ are unknown or difficult to keep track of, but can be best approximated by their empirical measure $\hat{\mu}_\ell^n(dx) = \sum_i \delta(x - x_\ell^i) dx / n$. Therefore, it is natural to think that $\phi_{\hat{\mu}_\ell^n, p}^*$, with $\mu$ in (5) replaced by $\hat{\mu}_\ell^n$, provides the best update direction for moving the particles (and equivalently $\hat{\mu}_\ell^n$) "closer to" $\nu_p$. Implementing this update (8) iteratively, we get the main SVGD algorithm in Algorithm 1.

Intuitively, the update in (8) pushes the particles towards the high probability regions of the target probability via the gradient term $\nabla \log p$, while maintaining a degree of diversity via the second term $\nabla k(x, x_i)$. In addition, (8) reduces to the typical gradient descent for maximizing $\log p$ if we use only a single particle ($n = 1$) and the kernel stratifies $\nabla k(x, x') = 0$ for $x = x'$; this allows SVGD to provide a spectrum of approximation that smooths between maximum a posterior (MAP) optimization to a full sampling approximation by using different particle sizes, enabling efficient trade-off between accuracy and computation cost.

Despite the similarity to gradient descent, we should point out that the SVGD update in (8) does not correspond to minimizing any objective function $F(\{x_\ell^i\})$ in terms of the particle location $\{x_\ell^i\}$, because one would find $\partial_{x_i} \partial_{x_j} F \neq \partial_{x_j} \partial_{x_i} F$ if this is true. Instead, it is best to view SVGD as a type of (particle-based) numerical approximation of an evolutionary partial differential equation (PDE) of densities or measures, which corresponds to a special type of gradient flow of the KL divergence functional whose equilibrium state equals the given target distribution $\nu_p$, as we discuss in the sequel.

## 3 Density Evolution of SVGD Dynamics

This section collects our main results. We characterize the evolutionary process of the empirical measures $\hat{\mu}_\ell^n$ of the SVGD particles and their large sample limit as $n \to \infty$ (Section 3.1) and large time limit as $\ell \to \infty$ (Section 3.2), which together establish the weak convergence of $\hat{\mu}_\ell^n$ to the target measure $\nu_p$. Further, we show that the large sample limit of the SVGD dynamics is characterized by a Vlasov process, which monotonically decreases the KL divergence to target distributions with a decreasing rate that equals the square of Stein discrepancy (Section 3.2-3.3). We also establish a geometric intuition that interpret SVGD as a gradient flow of KL divergence under a new Riemannian metric structure induced by Stein operator (Section 3.4). Section 3.5 provides a brief discussion on the connection to Langevin dynamics.

## 3.1 Large Sample Asymptotic of SVGD

Consider the optimal transform $\boldsymbol{T}_{\mu,p}(x) = x + \epsilon \boldsymbol{\phi}^*_{\mu,p}(x)$ with $\boldsymbol{\phi}^*_{\mu,p}$ defined in (5). We define its related map $\Phi_p \colon \mu \mapsto \boldsymbol{T}_{\mu,p}\mu$, where $\boldsymbol{T}_{\mu,p}\mu$ denotes the pushforward measure of $\mu$ through transform $\boldsymbol{T}_{\mu,p}$. This map fully characterizes the SVGD dynamics in the sense that the empirical measure $\hat{\mu}^n_\ell$ can be obtained by recursively applying $\Phi_p$ starting from the initial measure $\hat{\mu}^n_0$.

$$\hat{\mu}^n_{\ell+1} = \Phi_p(\hat{\mu}^n_\ell), \quad \forall \ell \in \mathbb{N}. \tag{9}$$

Note that $\Phi_p$ is a nonlinear map because the transform $\boldsymbol{T}_{\mu,p}$ depends on the input map $\mu$. If $\mu$ has a density $q$ and $\epsilon$ is small enough so that $\boldsymbol{T}_{\mu,p}$ is invertible, the density $q'$ of $\mu' = \Phi_p(\mu)$ is given by the change of variables formula:

$$q'(z) = q(\boldsymbol{T}^{-1}_{\mu,p}(z)) \cdot |\det(\nabla \boldsymbol{T}^{-1}_{\mu,p}(z))|. \tag{10}$$

When $\mu$ is an empirical measure and $q$ is a Dirac delta function, this equation still holds formally in the sense of distribution (generalized functions).

Critically, $\Phi_p$ also fully characterizes the large sample limit property of SVGD. Assume the initial empirical measure $\hat{\mu}^n_0$ at the 0-th iteration weakly converges to a measure $\mu^\infty_0$ as $n \to \infty$, which can be achieved, for example, by drawing $\{x^i_0\}$ i.i.d. from $\mu^\infty_0$, or using MCMC or Quasi Monte Carlo methods. Starting from the limit initial measure $\mu^\infty_0$ and applying $\Phi_p$ recursively, we get

$$\mu^\infty_{\ell+1} = \Phi_p(\mu^\infty_\ell), \quad \forall \ell \in \mathbb{N}. \tag{11}$$

Assuming $\hat{\mu}^n_0 \Rightarrow \mu^\infty_0$ by initialization, we may expect that $\hat{\mu}^n_\ell \Rightarrow \mu^\infty_\ell$ for all the finite iterations $\ell$ if $\Phi_p$ satisfies certain Lipschitz condition. This is naturally captured by the bounded Lipschitz metric.

For two measures $\mu$ and $\nu$, their bounded Lipschitz (BL) metric is defined to be their difference of means on the set of bounded, Lipschitz test functions:

$$\mathrm{BL}(\mu,\ \nu) = \sup_f \left\{ \mathbb{E}_\mu f - \mathbb{E}_\nu f \ \ s.t. \ ||f||_{\mathrm{BL}} \le 1 \right\}, \quad \text{where} \quad ||f||_{\mathrm{BL}} = \max\{||f||_\infty,\ ||f||_{\mathrm{Lip}}\},$$

where $||f||_\infty = \sup_x |f(x)|$ and $||f||_{\mathrm{Lip}} = \sup_{x \ne y} \frac{|f(x)-f(y)|}{||x-y||_2}$. For a vector-valued bounded Lipschitz function $\boldsymbol{f} = [f_1, \ldots, f_d]^\top$, we define its norm by $||\boldsymbol{f}||^2_{\mathrm{BL}} = \sum_{i=1}^d ||f_i||^2_{\mathrm{BL}}$. It is known that the BL metric metricizes weak convergence, that is, $\mathrm{BL}(\mu_n,\ \nu) \to 0$ if and only if $\mu_n \Rightarrow \nu$.

**Lemma 3.1.** *Assuming $\boldsymbol{g}(x,y) \coloneqq \mathcal{S}^x_p \otimes k(x,y)$ is bounded Lipschitz jointly on $(x,y)$ with norm $||\boldsymbol{g}||_{\mathrm{BL}} < \infty$, then for any two probability measures $\mu$ and $\mu'$, we have*

$$\mathrm{BL}(\Phi_p(\mu),\ \Phi_p(\mu')) \le (1 + 2\epsilon ||\boldsymbol{g}||_{\mathrm{BL}}) \, \mathrm{BL}(\mu,\ \mu').$$

**Theorem 3.2.** *Let $\hat{\mu}^n_\ell$ be the empirical measure of $\{x^i_\ell\}_{i=1}^n$ at the $\ell$-th iteration of SVGD. Assuming*

$$\lim_{n \to \infty} \mathrm{BL}(\hat{\mu}^n_0,\ \mu^\infty_0) \to 0,$$

*then for $\mu^\infty_\ell$ defined in* (11)*, at any finite iteration $\ell$, we have*

$$\lim_{n \to \infty} \mathrm{BL}(\hat{\mu}^n_\ell,\ \mu^\infty_\ell) \to 0.$$

*Proof.* It is a direct result of Lemma 3.1. $\qquad\qquad\qquad\qquad\qquad\qquad\qquad\qquad\qquad\qquad\square$

Since $\mathrm{BL}(\mu,\ \nu)$ metricizes weak convergence, our result suggests $\hat{\mu}^n_\ell \Rightarrow \hat{\mu}^\infty_\ell$ for $\forall \ell$, if $\hat{\mu}^n_0 \Rightarrow \hat{\mu}^\infty_0$ by initialization. The bound of BL metric in Lemma 3.1 increases by a factor of $(1 + 2\epsilon ||g||_{\mathrm{BL}})$ at each iteration. We can prevent the explosion of the BL bound by decaying step size sufficiently fast. It may be possible to obtain tighter bounds, however, it is fundamentally impossible to get a factor smaller than one without further assumptions: suppose we can get $\mathrm{BL}(\Phi_p(\mu),\ \Phi_p(\mu')) \le \alpha \mathrm{BL}(\mu,\ \mu')$ for some constant $\alpha \in [0,1)$, then starting from any initial $\hat{\mu}^n_0$, with any fixed particle size $n$ (e.g., $n = 1$), we would have $\mathrm{BL}(\hat{\mu}^n_\ell,\ \nu_p) = O(\alpha^\ell) \to 0$ as $\ell \to 0$, which is impossible because we can not get arbitrarily accurate approximate of $\nu_p$ with finite $n$. It turns out that we need to look at KL divergence in order to establish convergence towards $\nu_p$ as $\ell \to \infty$, as we discuss in Section 3.2-3.3.

**Remark** *Because $g(x, y) = \nabla_x \log p(x) k(x, y) + \nabla_x k(x, y)$, and $\nabla_x \log p(x)$ is often unbounded if the domain $X$ is not unbounded. Therefore, the condition that $g(x, y)$ must be bounded in Lemma 3.1 suggests that it can only be used when $X$ is compact. It is an open question to establish results that can work for more general domain $X$.*

## 3.2 Large Time Asymptotic of SVGD

Theorem 3.2 ensures that we only need to consider the update (11) starting from the limit initial $\mu_0^\infty$, which we can assume to have nice density functions and have finite KL divergence with the target $\nu_p$. We show that update (11) monotonically decreases the KL divergence between $\mu_\ell^\infty$ and $\nu_p$ and hence allows us to establish the convergence $\mu_\ell^\infty \Rightarrow \nu_p$.

**Theorem 3.3.** *1. Assuming $p$ is a density that satisfies Stein's identity (3) for $\forall \phi \in \mathcal{H}$, then the measure $\nu_p$ of $p$ is a fixed point of map $\Phi_p$ in (11).*

*2. Assume $R = \sup_x \{\frac{1}{2} ||\nabla \log p||_{\mathrm{Lip}} k(x, x) + 2\nabla_{xx'} k(x, x)\} < \infty$, where $\nabla_{xx'} k(x, x) = \sum_i \partial_{x_i} \partial_{x'_i} k(x, x')|_{x=x''}$, and the step size $\epsilon_\ell$ at the $\ell$-th iteration is no larger than $\epsilon_\ell^* := (2 \sup_x \rho(\nabla \phi_{\mu_\ell, p}^* + \nabla \phi_{\mu_\ell, p}^{*\top}))^{-1}$, where $\rho(A)$ denotes the spectrum norm of a matrix $A$. If $\mathrm{KL}(\mu_0^\infty \,||\, \nu_p) < \infty$ by initialization, then*

$$\frac{1}{\epsilon_\ell} \left[ \mathrm{KL}(\mu_{\ell+1}^\infty \,||\, \nu_p) - \mathrm{KL}(\mu_\ell^\infty \,||\, \nu_p) \right] \leq -(1 - \epsilon_\ell R) \, \mathbb{D}(\mu_\ell^\infty \,||\, \nu_p)^2, \tag{12}$$

*that is, the population SVGD dynamics always deceases the KL divergence when using sufficiently small step sizes, with a decreasing rate upper bounded by the squared Stein discrepancy. Further, if we set the step size $\epsilon_\ell$ to be $\epsilon_\ell \propto \mathbb{D}(\mu_\ell^\infty \,||\, \nu_p)^\beta$ for any $\beta > 0$, then (12) implies that $\mathbb{D}(\mu_\ell^\infty \,||\, \nu_p) \to 0$ as $\ell \to \infty$.*

**Remark** *Assuming $\mathbb{D}(\mu_\ell^\infty \,||\, \nu_p) \to 0$ implies $\mu_\ell^\infty \Rightarrow \nu_p$ (see (7)), then Theorem 3.3(2) implies $\mu_\ell^\infty \Rightarrow \nu_p$. Further, together with Theorem 3.2, we can establish the weak convergence of the empirical measures of the SVGD particles: $\hat{\mu}_\ell^n \Rightarrow \nu_p$, as $\ell \to \infty$, $n \to \infty$.*

**Remark** *Theorem 3.3 can not be directly applied on the empirical measures $\hat{\mu}_\ell^n$ with finite sample size $n$, since it would give $\mathrm{KL}(\hat{\mu}_\ell^n \,||\, \nu_p) = \infty$ in the beginning. It is necessary to use BL metric and KL divergence to establish the convergence w.r.t. sample size $n$ and iteration $\ell$, respectively.*

**Remark** *The requirement that $\epsilon_\ell \leq \epsilon_\ell^*$ is needed to guarantee that the transform $T_{\mu_\ell, p}(x) = x + \epsilon \phi_{\mu_\ell, p}^*(x)$ has a non-singular Jacobean matrix everywhere. From the bound in Equation A.6 of the Appendix, we can derive an upper bound of the spectrum radius:*

$$\sup_x \rho(\nabla \phi_{\mu_\ell, p}^* + \nabla \phi_{\mu_\ell, p}^{*\top}) \leq 2 \sup_x ||\nabla \phi_{\mu_\ell, p}^*||_F \leq 2 \sup_x \sqrt{\nabla_{xx'} k(x, x)} \, \mathbb{D}(\mu_\ell \,||\, \nu_p).$$

*This suggest that the step size should be upper bounded by the inverse of Stein discrepancy, i.e., $\epsilon_\ell^* \propto \mathbb{D}(\mu_\ell \,||\, \nu_p)^{-1} = ||\phi_{\mu_\ell, p}^*||_{\mathcal{H}}^{-1}$, where $\mathbb{D}(\mu_\ell \,||\, \nu_p)$ can be estimated using (6) (see [7]).*

## 3.3 Continuous Time Limit and Vlasov Process

Many properties can be understood more easily as we take the continuous time limit ($\epsilon \to 0$), reducing our system to a partial differential equation (PDE) of the particle densities (or measures), under which we show that the negative gradient of KL divergence exactly equals the square Stein discrepancy (the limit of (12) as $\epsilon \to 0$).

To be specific, we define a continuous time $t = \epsilon \ell$, and take infinitesimal step size $\epsilon \to 0$, the evolution of the density $q$ in (10) then formally reduces to the following nonlinear Fokker-Planck equation (see Appendix A.3 for the derivation):

$$\frac{\partial}{\partial t} q_t(x) = -\nabla \cdot (\phi_{q_t, p}^*(x) q_t(x)). \tag{13}$$

This PDE is a type of deterministic Fokker-Planck equation that characterizes the movement of particles under deterministic forces, but it is *nonlinear* in that the velocity field $\phi_{q_t, p}^*(x)$ depends on the current particle density $q_t$ through the drift term $\phi_{q_t, p}^*(x) = \mathbb{E}_{x' \sim q_t}[\mathcal{S}_p^{x'} \otimes k(x, x')]$.

It is not surprising to establish the following continuous version of Theorem 3.3(2), which is of central importance to our gradient flow perspective in Section 3.4:

**Theorem 3.4.** *Assuming* $\{\mu_t\}$ *are the probability measures whose densities* $\{q_t\}$ *satisfy the PDE in* (13), *and* $\mathrm{KL}(\mu_0 \parallel \nu_p) < \infty$, *then*

$$\frac{\mathrm{d}}{\mathrm{d}t}\mathrm{KL}(\mu_t \parallel \nu_p) = -\mathbb{D}(\mu_t \parallel \nu_p)^2. \tag{14}$$

**Remark** *This result suggests a path integration formula,* $\mathrm{KL}(\mu_0 \parallel \nu_p) = \int_0^\infty \mathbb{D}(\mu_t \parallel \nu_p)^2 \mathrm{d}t$, *which can be potentially useful for estimating KL divergence or the normalization constant.*

PDE (13) only works for differentiable densities $q_t$. Similar to the case of $\Phi_p$ as a map between (empirical) measures, one can extend (13) to a measure-value PDE that incorporates empirical measures as weak solutions. Take a differentiable test function $h$ and integrate the both sides of (13):

$$\int \frac{\partial}{\partial t} h(x) q_t(x) \mathrm{d}x = - \int h(x) \nabla \cdot (\phi_{q_t,p}^*(x) q_t(x)) \mathrm{d}x,$$

Using integration by parts on the right side to "shift" the derivative operator from $\phi_{q_t,p}^* q_t$ to $h$, we get

$$\frac{\mathrm{d}}{\mathrm{d}t}\mathbb{E}_{\mu_t}[h] = \mathbb{E}_{\mu_t}[\nabla h^\top \phi_{\mu_t,p}^*], \tag{15}$$

which depends on $\mu_t$ only through the expectation operator and hence works for empirical measures as well,. A set of measures $\{\mu_t\}$ is called the weak solution of (13) if it satisfies (15).

Using results in Fokker-Planck equation, the measure process (13)-(15) can be translated to an ordinary differential equation on random particles $\{x_t\}$ whose distribution is $\mu_t$:

$$\mathrm{d}x_t = \phi_{\mu_t,p}^*(x_t)\mathrm{d}t, \qquad \mu_t \text{ is the distribution of random variable } x_t, \tag{16}$$

initialized from random variable $x_0$ with distribution $\mu_0$. Here the nonlinearity is reflected in the fact that the velocity field depends on the distribution $\mu_t$ of the particle at the current time.

In particular, if we initialize (15) using an empirical measure $\hat{\mu}_0^n$ of a set of finite particles $\{x_0^i\}_{i=1}^n$, (16) reduces to the following continuous time limit of $n$-particle SVGD dynamics:

$$\mathrm{d}x_t^i = \phi_{\hat{\mu}_t^n,p}^*(x_t^i)\mathrm{d}t, \quad \forall i = 1, \dots, n, \qquad \text{with} \quad \hat{\mu}_t^n(\mathrm{d}x) = \frac{1}{n}\sum_{i=1}^n \delta(x - x_t^i)\mathrm{d}x, \tag{17}$$

where $\{\hat{\mu}_t^n\}$ can be shown to be a weak solution of (13)-(15), parallel to (9) in the discrete time case. (16) can be viewed as the large sample limit ($n \to \infty$) of (17).

The process (13)-(17) is a type of *Vlasov processes* [12, 13], which are (deterministic) interacting particle processes of the particles interacting with each other though the dependency on their "mean field" $\mu_t$ (or $\hat{\mu}_t^n$), and have found important applications in physics, biology and many other areas. There is a vast literature on theories and applications of interacting particles systems in general, and we only refer to Spohn [14], Del Moral [15] and references therein as examples. Our particular form of Vlasov process, constructed based on Stein operator in order to approximate arbitrary given distributions, seems to be new to the best of our knowledge.

### 3.4 Gradient Flow, Optimal Transport, Geometry

We develop a geometric view for the Vlasov process in Section 3.3, interpreting it as a gradient flow for minimizing the KL divergence functional, defined on a new type of optimal transport metric on the space of density functions induced by Stein operator.

We focus on the set of "nice" densities $q$ paired with a well defined Stein operator $\mathcal{S}_q$, acting on a Hilbert space $\mathcal{H}$. To develop the intuition, consider a density $q$ and its nearby density $q'$ obtained by applying transform $\boldsymbol{T}(x) = x + \boldsymbol{\phi}(x)\mathrm{d}t$ on $x \sim q$ with infinitesimal $\mathrm{d}t$ and $\boldsymbol{\phi} \in \mathcal{H}$, then we can show that (See Appendix A.3)

$$\log q'(x) = \log q(x) - \mathcal{S}_q\boldsymbol{\phi}(x)\mathrm{d}t, \qquad q'(x) = q(x) - q(x)\mathcal{S}_q\boldsymbol{\phi}(x)\mathrm{d}t, \tag{18}$$

Because one can show that $\mathcal{S}_q\phi = \frac{\nabla\cdot(\phi q)}{q}$ from (2), we define operator $q\mathcal{S}_q$ by $q\mathcal{S}_q\phi(x) = q(x)\mathcal{S}_q\phi(x) = \nabla\cdot(\phi(x)q(x))$. Eq (18) suggests that the Stein operator $\mathcal{S}_q$ (resp. $q\mathcal{S}_q$) serves to translate a $\phi$-perturbation on the random variable $x$ to the corresponding change on the log-density (resp. density). This fact plays a central role in our development.

Denote by $\mathcal{H}_q$ (resp. $q\mathcal{H}_q$) the space of functions of form $\mathcal{S}_q\phi$ (resp. $q\mathcal{S}_q\phi$) with $\phi \in \mathcal{H}$, that is,

$$\mathcal{H}_q = \{\mathcal{S}_q\phi : \phi \in \mathcal{H}\}, \qquad\qquad q\mathcal{H}_q = \{q\mathcal{S}_q\phi : \phi \in \mathcal{H}\}.$$

Equivalently, $q\mathcal{H}_q$ is the space of functions of form $qf$ where $f \in \mathcal{H}_q$. This allows us to consider the inverse of Stein operator for functions in $\mathcal{H}_q$. For each $f \in \mathcal{H}_q$, we can identify an unique function $\psi_f \in \mathcal{H}$ that has minimum $||\cdot||_{\mathcal{H}}$ norm in the set of $\psi$ that satisfy $\mathcal{S}_q\psi = f$, that is,

$$\psi_{q,f} = \arg\min_{\psi\in\mathcal{H}} \{||\psi||_{\mathcal{H}} \quad s.t. \quad \mathcal{S}_q\psi = f\},$$

where $\mathcal{S}_q\psi = f$ is known as the *Stein equation*. This allows us to define inner products on $\mathcal{H}_q$ and $q\mathcal{H}_q$ using the inner product on $\mathcal{H}$:

$$\langle f_1\ f_2\rangle_{\mathcal{H}_q} := \langle qf_1,\ qf_2\rangle_{q\mathcal{H}_q} := \langle \psi_{q,f_1},\ \psi_{q,f_2}\rangle_{\mathcal{H}}. \tag{19}$$

Based on standard results in RKHS [e.g., 16], one can show that if $\mathcal{H}$ is a RKHS with kernel $k(x, x')$, then $\mathcal{H}_q$ and $q\mathcal{H}_q$ are both RKHS; the reproducing kernel of $\mathcal{H}_q$ is $\kappa_p(x, x')$ in (6), and correspondingly, the kernel of $q\mathcal{H}_q$ is $q(x)\kappa_p(x, x')q(x')$.

Now consider $q$ and a nearby $q' = q + qf\mathrm{d}t, \forall f \in \mathcal{H}_q$, obtained by an infinitesimal perturbation on the density function using functions in space $\mathcal{H}_q$. Then the $\psi_{q,f}$ can be viewed as the "optimal" transform, in the sense of having minimum $||\cdot||_{\mathcal{H}}$ norm, that transports $q$ to $q'$ via $T(x) = x + \psi_{q,f}(x)\mathrm{d}t$. It is therefore natural to define a notion of distance between $q$ and $q' = q + qf\mathrm{d}t$ via,

$$\mathbb{W}_{\mathcal{H}}(q,\ q') := ||\psi_{q,f}||_{\mathcal{H}}\mathrm{d}t.$$

From (18) and (19), this is equivalent to

$$\mathbb{W}_{\mathcal{H}}(q,\ q') = ||q - q'||_{q\mathcal{H}_q}\mathrm{d}t = ||\log q' - \log q||_{\mathcal{H}_q}\mathrm{d}t.$$

Under this definition, we can see that the infinitesimal neighborhood $\{q' : \mathbb{W}_{\mathcal{H}}(q,\ q') \leq \mathrm{d}t\}$ of $q$, consists of densities (resp. log-densities) of form

$$q' = q + g\mathrm{d}t, \quad \forall g \in q\mathcal{H}_q,\ \ ||g||_{q\mathcal{H}_q} \leq 1,$$
$$\log q' = \log q + f\mathrm{d}t, \quad \forall f \in \mathcal{H}_q,\ \ ||f||_{\mathcal{H}_q} \leq 1.$$

Geometrically, this means that $q\mathcal{H}_q$ (resp. $\mathcal{H}_q$) can be viewed as the tangent space around density $q$ (resp. log-density $\log q$). Therefore, the related inner product $\langle\cdot,\ \cdot\rangle_{q\mathcal{H}_q}$ (resp. $\langle\cdot,\ \cdot\rangle_{\mathcal{H}_q}$) forms a Riemannian metric structure that corresponds to $\mathbb{W}_{\mathcal{H}}(q,\ q')$.

This also induces a geodesic distance that corresponds to a general, $\mathcal{H}$-dependent form of optimal transport metric between distributions. Consider two densities $p$ and $q$ that can be transformed from one to the other with functions in $\mathcal{H}$, in the sense that there exists a curve of velocity fields $\{\phi_t : \phi_t \in \mathcal{H},\ t \in [0,1]\}$ in $\mathcal{H}$, that transforms random variable $x_0 \sim q$ to $x_1 \sim p$ via $\mathrm{d}x_t = \phi_t(x)\mathrm{d}t$. This is equivalent to say that there exists a curve of densities $\{\rho_t : t \in [0,1]\}$ such that

$$\partial_t \rho_t = -\nabla\cdot(\phi_t\rho_t), \quad \text{and} \quad \rho_0 = q,\ \rho_1 = p.$$

It is therefore natural to define a geodesic distance between $q$ and $p$ via

$$\mathbb{W}_{\mathcal{H}}(q,\ p) = \inf_{\{\phi_t,\ \rho_t\}} \left\{ \int_0^1 ||\phi_t||_{\mathcal{H}}dt, \quad s.t. \quad \partial_t\rho_t = -\nabla\cdot(\phi_t\rho_t),\ \rho_0 = p,\ \rho_1 = q \right\}. \tag{20}$$

We call $\mathbb{W}_{\mathcal{H}}(p,q)$ an $\mathcal{H}$-Wasserstein (or optimal transport) distance between $p$ and $q$, in connection with the typical 2-Wasserstein distance, which can be viewed as a special case of (20) by taking $\mathcal{H}$ to be the $L^2_{\rho_t}$ space equipped with norm $||f||_{L^2_{\rho_t}} = \mathbb{E}_{\rho_t}[f^2]$, replacing the cost with $\int ||\phi_t||_{L^2_{\rho_t}}dt$; the 2-Wasserstein distance is widely known to relate to Langevin dynamics as we discuss more in Section 3.5 [e.g., 17, 18].

Now for a given functional $F(q)$, this metric structure induced a notion of functional covariant gradient: the covariant gradient $\mathrm{grad}_{\mathcal{H}}F(q)$ of $F(q)$ is defined to be a functional that maps $q$ to an element in the tangent space $q\mathcal{H}_q$ of $q$, and satisfies

$$F(q + f\mathrm{d}t) = F(q) + \langle\mathrm{grad}_{\mathcal{H}}F(q),\ f\mathrm{d}t\rangle_{q\mathcal{H}_q}, \tag{21}$$

for any $f$ in the tangent space $q\mathcal{H}_q$.

**Theorem 3.5.** *Following* (21)*, the gradient of the KL divergence functional* $F(q) := \mathrm{KL}(q \,||\, p)$ *is*

$$\mathrm{grad}_{\mathcal{H}}\mathrm{KL}(q \,||\, p) = \nabla \cdot (\boldsymbol{\phi}^*_{q,p} q).$$

*Therefore, the SVGD-Valsov equation* (13) *is a gradient flow of KL divergence under metric* $\mathbb{W}_{\mathcal{H}}(\cdot, \cdot)$:

$$\frac{\partial q_t}{\partial t} = -\mathrm{grad}_{\mathcal{H}}\mathrm{KL}(q_t \,||\, p).$$

*In addition,* $||\mathrm{grad}_{\mathcal{H}}\mathrm{KL}(q \,||\, p)||_{q\mathcal{H}_q} = \mathbb{D}(q \,||\, p)$.

**Remark** *We can also definite the functional gradient via*

$$\mathrm{grad}_{\mathcal{H}}F(q) \propto \underset{f:\, ||f||_{q\mathcal{H}_q} \leq 1}{\arg\max} \left\{ \lim_{\epsilon \to 0^+} \frac{F(q + \epsilon f) - F(q)}{\mathbb{W}_{\mathcal{H}}(q + \epsilon f,\, q)} \right\},$$

*which specifies the steepest ascent direction of* $F(q)$ *(with unit norm). The result in Theorem* (3.5) *is consistent with this definition.*

### 3.5 Comparison with Langevin Dynamics

The theory of SVGD is parallel to that of Langevin dynamics in many perspectives, but with importance differences. We give a brief discussion on their similarities and differences.

Langevin dynamics works by iterative updates of form

$$x_{\ell+1} \leftarrow x_\ell + \epsilon \nabla \log p(x_\ell) + 2\sqrt{\epsilon}\xi_\ell, \quad \xi_\ell \sim \mathcal{N}(0, 1),$$

where a *single* particle $\{x_\ell\}$ moves along the gradient direction, perturbed with a random Gaussian noise that plays the role of enforcing the diversity to match the variation in $p$ (which is accounted by the deterministic repulsive force in SVGD). Taking the continuous time limit ($\epsilon \to 0$), We obtain a Ito stochastic differential equation, $\mathrm{d}x_t = -\nabla \log p(x_t)\mathrm{d}t + 2\mathrm{d}W_t$,where $W_t$ is a standard Brownian motion, and $x_0$ is a random variable with initial distribution $q_0$. Standard results show that the density $q_t$ of random variable $x_t$ is governed by a *linear* Fokker-Planck equation, following which the KL divergence to $p$ decreases with a rate that equals Fisher divergence:

$$\frac{\partial q_t}{\partial t} = -\nabla \cdot (q_t \nabla \log p) + \Delta q_t, \qquad \frac{\mathrm{d}}{\mathrm{d}t}\mathrm{KL}(q_t \,||\, p) = -\mathbb{F}(q_t, p), \qquad (22)$$

where $\mathbb{F}(q, p) = ||\nabla \log(q/p)||^2_{L^2_q}$. This result is parallel to Theorem 3.4, and the role of square Stein discrepancy (and RKHS $\mathcal{H}$) is replaced by Fisher divergence (and $L^2_q$ space). Further, parallel to Theorem 3.5, it is well known that (22) can be also treated as a gradient flow of the KL functional $\mathrm{KL}(q \,||\, p)$, but under the 2-Wasserstein metric $\mathbb{W}_2(q,\, p)$ [17]. The main advantage of using RKHS over $L^2_q$ is that it allows tractable computation of the optimal transport direction; this is not case when using $L^2_q$ and as a result Langevin dynamics requires a random diffusion term in order to form a proper approximation.

Practically, SVGD has the advantage of being deterministic, and reduces to exact MAP optimization when using only a single particle, while Langevin dynamics has the advantage of being a standard MCMC method, inheriting its statistical properties, and does not require an $O(n^2)$ cost to calculate the $n$-body interactions as SVGD. However, the connections between SVGD and Langevin dynamics may allow us to develop theories and algorithms that unify the two, or combine their advantages.

## 4 Conclusion and Open Questions

We developed a theoretical framework for analyzing the asymptotic properties of Stein variational gradient descent. Many components of the analysis provide new insights in both theoretical and practical aspects. For example, our new metric structure can be useful for solving other learning problems by leveraging its computational tractability. Many important problems remains to be open. For example, an important open problem is to establish explicit convergence rate of SVGD, for which the existing theoretical literature on Langevin dynamics and interacting particles systems may provide insights. Another problem is to develop finite sample bounds for SVGD that can take the fact that it reduces to MAP optimization when $n = 1$ into account. It is also an important direction to understand the bias and variance of SVGD particles, or combine it with traditional Monte Carlo whose bias-variance analysis is clearer (see e.g., [19]).

**Acknowledgement**    This work is supported in part by NSF CRII 1565796. We thank Lester Mackey and the anonymous reviewers for their comments.

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
