[Supplementary Material]

# Appendix of "Stein Variational Gradient Descent as Gradient Flow"

**Qiang Liu**
Department of Computer Science
Dartmouth College
Hanover, NH 03755
qiang.liu@dartmouth.edu

## A  Density Evolution of SVGD Dynamics

### A.1  Proof of Lemma 3.1

*Proof.* Recall that $\boldsymbol{g}(x, x') = \mathcal{S}_p^{x'} \otimes k(x', x)$, and $\boldsymbol{\phi}_{\mu,p}^*(x) = \mathbb{E}_{x' \sim \mu}[\boldsymbol{g}(x, x')]$, we have $\boldsymbol{T}_{\mu,p}(x) = x + \epsilon \mathbb{E}_{x' \sim \mu}[\boldsymbol{g}(x, x')]$. Therefore,

$$
\begin{aligned}
||\boldsymbol{T}_{\mu,p}||_{\mathrm{Lip}} &= \max_{x \neq y} \frac{||\boldsymbol{T}_{\mu,p}(x) - \boldsymbol{T}_{\mu,p}(y)||_2}{||x - y||_2} \\
&= \max_{x \neq y} \frac{||x - y + \epsilon \mathbb{E}_{x' \sim \mu}[\boldsymbol{g}(x, x') - \boldsymbol{g}(y, x')]||_2}{||x - y||_2} \\
&\leq 1 + \epsilon ||\boldsymbol{g}||_{\mathrm{Lip}},
\end{aligned} \tag{A.1}
$$

and for $\forall\, x$,

$$
||\boldsymbol{T}_{\mu,p}(x) - \boldsymbol{T}_{\nu,p}(x)||_2 = \epsilon ||\mathbb{E}_{x' \sim \mu} \boldsymbol{g}(x, x') - \mathbb{E}_{x' \sim \nu} \boldsymbol{g}(x, x')||_2 \leq \epsilon ||\boldsymbol{g}||_{\mathrm{BL}} \, \mathrm{BL}(\mu, \nu). \tag{A.2}
$$

For any $h$ with $||h||_{\mathrm{BL}} = \max(||h||_\infty, ||h||_{\mathrm{Lip}}) \leq 1$, we have

$$
\begin{aligned}
&\big| \mathbb{E}_{\Phi_p(\mu)}[h] - \mathbb{E}_{\Phi_p(\nu)}[h] \big| \\
&= \big| \mathbb{E}_\mu[h \circ \boldsymbol{T}_{\mu,p}] - \mathbb{E}_\nu[h \circ \boldsymbol{T}_{\nu,p}] \big| \\
&\leq \big| \mathbb{E}_\mu[h \circ \boldsymbol{T}_{\mu,p}] - \mathbb{E}_\nu[h \circ \boldsymbol{T}_{\mu,p}] \big| + \big| \mathbb{E}_\nu[h \circ \boldsymbol{T}_{\mu,p}] - \mathbb{E}_\nu[h \circ \boldsymbol{T}_{\nu,p}] \big|.
\end{aligned}
$$

We just need to bound these two terms. For the first term,

$$
\begin{aligned}
\big| \mathbb{E}_\mu[h \circ \boldsymbol{T}_{\mu,p}] - \mathbb{E}_\nu[h \circ \boldsymbol{T}_{\mu,p}] \big| &\leq ||h \circ \boldsymbol{T}_{\mu,p}||_{\mathrm{BL}} \, \mathrm{BL}(\mu, \nu) \\
&\leq \max\big( ||h||_\infty, ||h||_{\mathrm{Lip}} ||\boldsymbol{T}_{\mu,p}||_{\mathrm{Lip}} \big) \, \mathrm{BL}(\mu, \nu) \\
&\leq (1 + \epsilon ||\boldsymbol{g}||_{\mathrm{Lip}}) \mathrm{BL}(\mu, \nu), \qquad \text{//by Equation A.1.}
\end{aligned}
$$

For the second term,

$$
\begin{aligned}
\big| \mathbb{E}_\nu[h \circ \boldsymbol{T}_{\mu,p}] - \mathbb{E}_\nu[h \circ \boldsymbol{T}_{\nu,p}] \big| &\leq \max_x \big| h \circ \boldsymbol{T}_{\mu,p}(x) - h \circ \boldsymbol{T}_{\nu,p}(x) \big| \\
&\leq ||h||_{\mathrm{Lip}} \max_x ||\boldsymbol{T}_{\mu,p}(x) - \boldsymbol{T}_{\nu,p}(x)||_2 \\
&\leq \epsilon ||\boldsymbol{g}||_{\mathrm{BL}} \, \mathrm{BL}(\mu, \nu), \qquad \text{//by Equation A.2.}
\end{aligned}
$$

Therefore,

$$
\mathrm{BL}(\Phi_p(\mu), \Phi_p(\nu)) \leq (1 + \epsilon ||\boldsymbol{g}||_{\mathrm{Lip}} + \epsilon ||\boldsymbol{g}||_{\mathrm{BL}}) \, \mathrm{BL}(\mu, \nu) \leq (1 + 2\epsilon ||\boldsymbol{g}||_{\mathrm{BL}}) \, \mathrm{BL}(\mu, \nu).
$$

$\square$

## A.2 Proof of Theorem 3.3

*Proof.* Denote by $\mu_\ell = \mu_\ell^\infty$ for notation convenience.

$$\mathrm{KL}(\mu_{\ell+1} \,||\, \nu_p) - \mathrm{KL}(\mu_\ell \,||\, \nu_p)$$

$$= \mathrm{KL}(\boldsymbol{T}_{\mu_\ell,p}\mu_\ell \,||\, \nu_p) - \mathrm{KL}(\mu_\ell \,||\, \nu_p)$$

$$= \mathrm{KL}(\mu_\ell \,||\, \boldsymbol{T}_{\mu_\ell,p}^{-1}\nu_p) - \mathrm{KL}(\mu_\ell \,||\, \nu_p) \qquad \text{{\color{magenta}//by Lemma A.2}}$$

$$= -\mathbb{E}_{x\sim\mu_\ell}[\log p(\boldsymbol{T}_{\mu_\ell,p}(x)) + \log\det(\nabla\boldsymbol{T}_{\mu_\ell,p}(x)) - \log p(x)]. \qquad (A.3)$$

Note that $\boldsymbol{T}_{\mu_\ell,p}(x) = x + \epsilon\boldsymbol{\phi}^*_{\mu_\ell,p}(x)$. We have the follow version of Taylor approximation:

$$\log p(x) - \log p(\boldsymbol{T}_{\mu_\ell,p}(x)) \leq -\epsilon\nabla_x\log p(x)^\top\boldsymbol{\phi}^*_{\mu_\ell,p}(x) \;+\; \frac{\epsilon^2}{2}||\nabla\log p||_{\mathrm{Lip}}\cdot||\boldsymbol{\phi}^*_{\mu_\ell,p}||_2^2. \quad (A.4)$$

This is because, defining $x_s = x + s\epsilon\boldsymbol{\phi}^*_{\mu_\ell,p}(x), \forall s \in [0,1]$,

$$\log p(x) - \log p(\boldsymbol{T}_{\mu_\ell,p}(x))$$

$$= -\int_0^1 \nabla_s\log p(x_s)\mathrm{d}s$$

$$= -\int_0^1 \nabla_x\log p(x_s)^\top(\epsilon\boldsymbol{\phi}^*_{\mu_\ell,p}(x))\,\mathrm{d}s$$

$$= -\epsilon\nabla_x\log p(x)^\top\boldsymbol{\phi}^*_{\mu_\ell,p}(x) \;-\; \int_0^1 (\nabla_x\log p(x_s) - \nabla_x\log p(x))^\top(\epsilon\boldsymbol{\phi}^*_{\mu_\ell,p}(x))\mathrm{d}s$$

$$\leq -\epsilon\nabla_x\log p(x)^\top\boldsymbol{\phi}^*_{\mu_\ell,p}(x) \;+\; \epsilon^2||\nabla\log p||_{\mathrm{Lip}}\cdot||\boldsymbol{\phi}^*_{\mu_\ell,p}(x)||_2^2\int_0^1 s\mathrm{d}s$$

$$= -\epsilon\nabla_x\log p(x)^\top\boldsymbol{\phi}^*_{\mu_\ell,p}(x) \;+\; \frac{\epsilon^2}{2}||\nabla\log p||_{\mathrm{Lip}}\cdot||\boldsymbol{\phi}^*_{\mu_\ell,p}(x)||_2^2.$$

where we used the fundamental theorem of calculus, which holds for weakly differentiable functions [20, Theorem 3.60, page 77]. In addition, Take $B = \nabla\boldsymbol{\phi}^*_{\mu_\ell,p}(x)$ in bound (A.9) of Lemma A.1, and take $\epsilon < 1/(2\rho(B+B^\top))$, we have

$$\log|\det(\nabla\boldsymbol{T}_{\mu_\ell,p}(x))| \geq \epsilon\,\mathrm{tr}(\nabla\boldsymbol{\phi}^*_{\mu_\ell,p}(x)) - 2\epsilon^2||\nabla\boldsymbol{\phi}^*_{\mu_\ell,p}(x)||_F^2$$

$$= \epsilon\,\nabla\cdot\boldsymbol{\phi}^*_{\mu_\ell,p}(x) - 2\epsilon^2||\nabla\boldsymbol{\phi}^*_{\mu_\ell,p}(x)||_F^2. \qquad (A.5)$$

Combining (A.4) and (A.5) gives

$$\mathrm{KL}(\mu_{\ell+1} \,||\, \nu_p) - \mathrm{KL}(\mu_\ell \,||\, \nu_p) \leq -\epsilon\mathbb{E}_{\mu_\ell}[\mathcal{S}_p\boldsymbol{\phi}^*_{\mu_\ell,p}] \;+\; \Delta$$

$$= -\epsilon\mathbb{D}(\mu_\ell \,||\, \nu_p)^2 \;+\; \Delta,$$

where $\Delta$ is a residual term:

$$\Delta = \epsilon^2\mathbb{E}_{x\sim\mu_\ell}\left[\frac{1}{2}||\nabla\log p||_{\mathrm{Lip}}\cdot||\boldsymbol{\phi}^*_{\mu_\ell,p}(x)||_2^2 + 2||\nabla\boldsymbol{\phi}^*_{\mu_\ell,p}(x)||_F^2\right]$$

We need to bound $||\boldsymbol{\phi}^*_{\mu_\ell,p}(x)||_2$ and $||\nabla\boldsymbol{\phi}^*_{\mu_\ell,p}(x)||_F$. This can be done using the reproducing property: let $\boldsymbol{\phi}^*_{\mu_\ell,p} = [\phi_1, \cdots, \phi_d]^\top$; recall that $\phi_i \in \mathcal{H}_0$ and $\boldsymbol{\phi}^*_{\mu_\ell,p} \in \mathcal{H} = \mathcal{H}_0^d$, then

$$\phi_i(x) = \langle\phi_i(\cdot),\, k(x,\cdot)\rangle_{\mathcal{H}_0}, \quad \partial_{x_j}\phi_i(x) = \langle\phi_i(\cdot),\, \partial_{x_j}k(x,\cdot)\rangle_{\mathcal{H}_0}, \qquad \forall i,j = 1,\ldots,d, \;\; x \in X.$$

Also note that $||\boldsymbol{\phi}^*_{\mu_\ell,p}||_{\mathcal{H}}^2 = \sum_{i=1}^d ||\phi_i||_{\mathcal{H}_0}^2 = \mathbb{D}(\mu_\ell \,||\, \nu_p)^2$, we have by Cauchy-Swarchz inequality,

$$||\boldsymbol{\phi}^*_{\mu_\ell,p}(x)||_2^2 = \sum_{i=1}^d \phi_i(x)^2$$

$$= \sum_{i=1}^d (\langle k(x,\cdot),\, \phi_i(\cdot)\rangle_{\mathcal{H}_0})^2$$

$$\leq \sum_i ||k(x,\cdot)||_{\mathcal{H}_0}^2 \cdot ||\phi_i||_{\mathcal{H}_0}^2$$

$$= k(x,x)\cdot||\boldsymbol{\phi}^*_{\mu_\ell,p}||_{\mathcal{H}}^2$$

$$= k(x,x)\cdot\mathbb{D}(\mu_\ell \,||\, \nu_p)^2,$$

and

$$\|\nabla \phi^*_{\mu_\ell,p}(x)\|^2_F = \sum_{ij} \partial_{x_j} \phi_i(x)^2$$

$$= \sum_{ij} (\langle \partial_{x_j} k(x,\cdot), \ \phi_i(\cdot)\rangle_{\mathcal{H}_0})^2$$

$$\leq \sum_{ij} \|\partial_{x_j} k(x,\cdot)\|^2_{\mathcal{H}_0} \cdot \|\phi_i\|^2_{\mathcal{H}_0}$$

$$= \sum_{ij} \partial_{x_j,x'_j} k(x,x')|_{x=x'} \cdot \|\phi_i\|^2_{\mathcal{H}_0}$$

$$= \nabla_{xx'} k(x,x) \cdot \|\phi^*_{\mu_\ell,p}\|^2_{\mathcal{H}}$$

$$= \nabla_{xx'} k(x,x) \cdot \mathbb{D}(\mu_\ell \ || \ \nu_p)^2. \tag{A.6}$$

Therefore,

$$\Delta \leq \epsilon^2 \, \mathbb{D}(\mu_\ell \ || \ \nu_p)^2 \left( \frac{1}{2} \mathbb{E}_{x \sim \mu_\ell}[\|\nabla \log p\|_{\text{Lip}} k(x,x) + 2\nabla_{xx'} k(x,x)] \right)$$

$$= \epsilon^2 R \, \mathbb{D}(\mu_\ell \ || \ \nu_p)^2.$$

This gives

$$\text{KL}(\mu_{\ell+1} \ || \ \nu_p) - \text{KL}(\mu_\ell \ || \ \nu_p) \leq -\epsilon \, (1 - \epsilon R) \, \mathbb{D}(\mu_\ell \ || \ \nu_p)^2.$$

$$\square$$

**Lemma A.1.** *Let $B$ be a square matrix and $\|B\|_F = \sqrt{\sum_{ij} b^2_{ij}}$ its Frobenius norm. Let $\epsilon$ be a positive number that satisfies $0 \leq \epsilon < \frac{1}{\rho(B+B^\top)}$, where $\rho(\cdot)$ denotes the spectrum radius. Then $I + \epsilon(B + B^\top)$ is positive definite, and*

$$\log |\det(I + \epsilon B)| \geq \epsilon \text{tr}(B) - \epsilon^2 \frac{\|B\|^2_F}{1 - \epsilon\rho(B + B^\top)}. \tag{A.7}$$

*Therefore, take an even smaller $\epsilon$ such that $0 \leq \epsilon \leq \frac{1}{2\rho(B+B^\top)}$, we get*

$$\log |\det(I + \epsilon B)| \geq \epsilon \text{tr}(B) - 2\epsilon^2 \|B\|^2_F. \tag{A.8}$$

*Proof.* When $\epsilon < \frac{1}{\rho(B+B^\top)}$, we have $\rho(I + \epsilon(B + B^\top)) \geq 1 - \epsilon\rho(B + B^\top) > 0$, so $I + \epsilon(B + B^\top)$ is positive definite.

By the property of matrix determinant, we have

$$\log |\det(I + \epsilon B)| = \frac{1}{2} \log \det((I + \epsilon B)(I + \epsilon B)^\top)$$

$$= \frac{1}{2} \log \det(I + \epsilon(B + B^\top) + \epsilon^2 BB^\top)$$

$$\geq \frac{1}{2} \log \det(I + \epsilon(B + B^\top)), \tag{A.9}$$

where (A.10) holds because both $I + \epsilon(B + B^\top)$ and $\epsilon^2 BB^\top$ are positive semi-definite.

Let $A = B + B^\top$. We can establish

$$\log \det(I + \epsilon A) \geq \epsilon \text{tr}(A) - \frac{\epsilon^2}{2} \frac{\|A\|^2_F}{1 - \epsilon\rho(A)}, \tag{A.10}$$

which holds for any symmetric matrix $A$ and $0 \leq \epsilon < 1/\rho(A)$. This is because, assuming $\{\lambda_i\}$ are the eigenvalues of $A$,

$$\log \det(I + \epsilon A) - \epsilon \mathrm{tr}(A) = \sum_i [\log(1 + \epsilon \lambda_i) - \epsilon \lambda_i]$$

$$= \sum_i \left[ \int_0^1 \frac{\epsilon \lambda_i}{1 + s\epsilon \lambda_i} \mathrm{d}s - \epsilon \lambda_i \right]$$

$$= -\sum_i \int_0^1 \frac{s\epsilon^2 \lambda_i^2}{1 + s\epsilon \lambda_i} \mathrm{d}s$$

$$\geq -\sum_i \frac{\epsilon^2 \lambda_i^2}{1 - \epsilon \max_i |\lambda_i|} \int_0^1 s \mathrm{d}s$$

$$\geq -\sum_i \frac{\epsilon^2 \lambda_i^2}{2(1 - \epsilon \max_i |\lambda_i|)}$$

$$= -\frac{\epsilon^2}{2} \frac{||A||_F^2}{1 - \epsilon \rho(A)}.$$

Take $A = B + B^\top$ in (A.11) and combine it with (A.10), we get

$$\log |\det(I + \epsilon B)| \geq \frac{1}{2} \log \det(I + \epsilon(B + B^\top))$$

$$\geq \frac{\epsilon}{2} \mathrm{tr}(B + B^\top) - \frac{\epsilon^2}{4} \frac{||B + B^\top||_F^2}{1 - \epsilon \rho(B + B^\top)}$$

$$\geq \epsilon \mathrm{tr}(B) - \epsilon^2 \frac{||B||_F^2}{1 - \epsilon \rho(B + B^\top)},$$

where we used the fact that $\mathrm{tr}(B) = \mathrm{tr}(B^\top)$ and $||B + B^\top||_F \leq ||B||_F + ||B^\top||_F = 2||B||_F$.  □

**Lemma A.2.** *Let $\boldsymbol{T}$ be a one-to-one map, and $\mu$ and $\nu$ two probability measures. We have*

$$\mathrm{KL}(\boldsymbol{T}\mu \mid\mid \nu) = \mathrm{KL}(\mu \mid\mid \nu),$$

*given that the KL divergence between $\mu$ and $\nu$ exists.*

*Proof.* We prove this for $f$-divergence in general, which includes KL divergence as a special case. Given a convex function $f$ such that $f(1) = 0$, the $f$-divergence is defined

$$\mathrm{D}_f(\mu \mid\mid \nu) = \mathbb{E}_\nu[f(\frac{\mathrm{d}\mu}{\mathrm{d}\nu})].$$

Assume $f^*$ is the convex conjugate of $f$, we have a variational representation for $f$-divergence:

$$\mathrm{D}_f(\mu \mid\mid \nu) = \sup_g \left\{ \mathbb{E}_\mu[g(x)] - \mathbb{E}_\nu[f^*(g(x))] \right\},$$

where $g$ is over the set of all measurable functions. Therefore, we have

$$\mathrm{D}_f(\boldsymbol{T}\mu \mid\mid \nu) = \sup_g \left\{ \mathbb{E}_\mu[g \circ T(x)] - \mathbb{E}_\nu[f^*(g(x))] \right\}$$

$$= \sup_{\tilde{g}} \left\{ \mathbb{E}_\mu[\tilde{g}(x)] - \mathbb{E}_\nu[f^*(\tilde{g} \circ \boldsymbol{T}^{-1}(x))] \right\} \qquad //\text{Define } \tilde{g} = g \circ \boldsymbol{T}.$$

$$= \mathrm{D}_f(\mu \mid\mid \boldsymbol{T}^{-1}\nu).$$

□

### A.3   Proof of Fokker-Planck Equation (13)

*Proof.* Recall that $\boldsymbol{T}_{\mu,p}(x) = x + \epsilon \phi_{\mu,p}^*(x)$ and we denote by $q$ the density of measure $\mu$. Assume $\epsilon$ is sufficiently small so that $\nabla \boldsymbol{T}_{\mu_p}(x) = I + \epsilon \nabla \phi_{\mu,p}^*(x)$ is positive definite (See Lemma A.1). By the implicit function theorem, we have

$$\boldsymbol{T}_{\mu,p}^{-1}(x) = x - \epsilon \phi_{\mu,p}^*(x) + o(\epsilon).$$

Therefore We have

$$\log q'(x) = \log q(\boldsymbol{T}_{\mu,p}^{-1}(x)) + \log \det(\nabla_x \boldsymbol{T}_{\mu,p}^{-1}(x))$$
$$= \log q(x - \epsilon \cdot \boldsymbol{\phi}_{\mu,p}^*(x)) + \log \det(I - \epsilon \nabla_x \boldsymbol{\phi}_{\mu,p}^*(x)) + o(\epsilon)$$
$$= \log q(x) - \epsilon \nabla_{x_i} \log q(x)^\top \boldsymbol{\phi}_{\mu,p}^*(x) - \epsilon q(x) \cdot \mathrm{tr}(\nabla_x \boldsymbol{\phi}_{\mu,p}^*(x)) + o(\epsilon)$$
$$= \log q(x) - \epsilon \mathcal{S}_q \boldsymbol{\phi}_{\mu,p}^*(x) + o(\epsilon).$$

Therefore,

$$\frac{q'(x) - q(x)}{\epsilon} = \frac{q(\log q(x) - \log q(x))}{\epsilon} + o(\epsilon)$$
$$= -q(x)\mathcal{S}_q \boldsymbol{\phi}_{q_\ell,p}^*(x) + o(\epsilon)$$
$$= -\nabla \cdot (\boldsymbol{\phi}_{q_\ell,p}^*(x) q_\ell(x)) + o(\epsilon).$$

Taking $\epsilon \to 0$ gives the result. $\square$

### A.4 Proof of Theorem 3.5

*Proof.* Since $q' = q + qf \mathrm{d}t$ is equivalent to transforming the variable by $\boldsymbol{T}(x) = x + \boldsymbol{\psi}_{q,f}\mathrm{d}t$, the corresponding change on KL divergence is

$$F(q + qf\mathrm{d}t) = F(q) + \mathbb{E}_q[\mathcal{S}_p \boldsymbol{\psi}_{q,f}]\mathrm{d}t$$
$$= F(q) + \langle \boldsymbol{\phi}_{q,p}^*, \ \boldsymbol{\psi}_{q,f} \rangle_{\mathcal{H}}\mathrm{d}t$$
$$= F(q) + \langle \nabla \cdot (\boldsymbol{\phi}_{q,p}^* q), \ \nabla \cdot (\boldsymbol{\psi}_{q,f} q) \rangle_{q\mathcal{H}_q}\mathrm{d}t$$
$$= F(q) + \langle \nabla \cdot (\boldsymbol{\phi}_{q,p}^* q), \ qf \rangle_{q\mathcal{H}_q}\mathrm{d}t$$

This proves that $\nabla \cdot (\boldsymbol{\phi}_{q,p}^* q)$ is the covariant functional gradient. $\square$