[Reviews · NeurIPS 2017]

Reviewer 1



–––RESPONSE TO AUTHOR REBUTTAL–––– The author's response is very thoughtful, but unfortunately it doesn't fully address my concerns. The fact that the proof of Lemma 3.1 works in the compact case is fine as far as it goes, but substantially detracts from how interesting the result is. The proposed condition on E_{y\sim \mu}[max_x(g(x,y))] is not sufficient since the same condition also has to hold for mu', which is a much more challenging condition to guarantee uniformly. It seems such a condition is also necessary in a Wasserstein metric version of Lemma 3.1. So the fix is non-trivial. The changes to Theorem 3.3 are more straightforward. Another issue which I forgot to include in my review is that the empirical measure converges in BL at a n^{-1/d} + n^{-1/2} rate, not the n^{-1/2} rate claimed at line 133. See [1] and [2]. Overall the paper requires substantial revision as well as some new technical arguments. I still recommend against acceptance. [1] Sriperumbudur et al. On the empirical estimation of integral probability metrics. Electron. J. Statist. Volume 6 (2012), 1550-1599. [2] Fournier and Guillin. On the rate of convergence in Wasserstein distance of the empirical measure. Probability Theory and Related Fields. Volume 162, Issue 3–4 (2015), pp 707–738. –––ORIGINAL REVIEW––– Summary: The paper investigates theoretical properties of Stein variational gradient descent (SVGD). The paper provides asymptotic convergence results, both in the large-particle and large-time limits. The paper also investigates the continuous-time limit of SVGD, which results in a PDE that has the flavor of a deterministic Fokker-Planck equation. Finally, the paper offers a geometric perspective, interpreting the continuous-time process as a gradient flow and introducing a novel optimal transport metric along the way. Overall, this is a very nice paper with some insightful results. However, there are a few important technical issues that prevent me from recommending publication. 1. Theorem 3.2: Via Lemma 3.1, Theorem 3.2 assumes g(x,y) is bounded and Lipschitz. However, if p is distantly dissipative, then g(x,y) is not bounded. Thus, the result as stated is essentially vacuous. That said, examining the proof, it should be straightforward to replace all of the bounded Lipschitz norms with Lipschitz norms and then get a result in terms of Wasserstein distance while only assuming g(x, y) is Lipschitz. 2. On a related note, the fact that the paper assumes eq. (7) holds “without further examination” (lines 78-79) seems ill-advised. As can be seen from the case of Theorem 3.2, it is important to validate that the hypotheses of a result actually hold and don’t interact badly with other (implicit) assumptions. The author(s) should more thoroughly discuss and show that the hypotheses in the two main theorems actually hold in some non-trivial cases. 3. Theorem 3.3: The final statement of the theorem, that eq. (12) implies the stein discrepancy goes to zero as \ell -> infinity, is not true. In particular, if \epsilon_\ell^* goes to zero too quickly, there’s no need for the Stein discrepancy to go to zero. Again, I think this statement can also be shown rigorously using the upper bound on the spectral radius discussed in the first remark after Thm 3.3. However, this should be worked out by the author(s). Minor comments: - \rho is never defined in Thm 3.3 - In many results and remarks: instead of writing “Assume …, then …”, one should writing either “Assuming …, then …” or “If …, then …”. - There were numerous other grammatical errors - \phi_{\mu,p}^* was used inconsistently: sometimes it was normalized to have norm 1, sometimes it wasn’t. - The references need to be updated: many of the papers that are only cited as being on the arXiv have now been published

Reviewer 2



This paper presents an analysis of SVGD algorithms. The main results are (1) a proof of the weak convergence of measures in the large sample, then large time limits, (2) characterization of the continuous time limit, (3) an interpretation of this continuous time process as optimizing an optimal transport metric . I am not qualified to assess many aspects of this paper, but from my understanding it is a strong contribution to a broadly important topic: approximation of distributions with particles. I particularly enjoyed the promise of relating/unifying these ideas to Langevin dynamics. My only question about the contribution is the applicability of the convergence results to practice. In particular, the questions regard the order of limit operations (N before L or vice versa). My questions are: (1) These limits don't apparently commute. Convergence in L of finite N measures, followed by convergence in N (lim N (lim L)) won't necessarily produce the same result as convergence in N of finite L measures, followed by convergence in L (lim L (lim N)). Is that true or did I misunderstand? This seems like an important point, I would recommend the authors clarify this distinction (if there is one) in the draft. Particularly Remark on lines 159-161. (2) It seems that practitioners might prefer (lim N (lim L)) to avoid storage issues. Can the authors comment on whether such results are possible? Interesting? Minor Typos: - "and hence and hence" line 149-150. - "It is not supervising" line 178

Reviewer 3



The authors provide a more detailed mathematical analysis of previous work on Stein variational gradient descent. In concert with other recent results, they prove consistency both in the discrete approximation and in the gradient descent. They also describe a continuous flow version of the method, as well as a geometric perspective. The consistency results bolster theoretical support for an existing method, and the latter two perspectives seem to serve mostly to set the stage for future work. I think the Stein operator perspective on variational inference is exciting and new, and this paper fills in some important theoretical gaps. Although it is not clear whether this work has immediate practical consequences, I still believe it represents a significant enough contribution to justify publication. Just a few minor typos: 150: "establish the converge" 155: rho has not been defined yet 168: "Many properties can be easier understood" 176: "velocity filed" 178: "It is not supervising to establish" 195: "continuos"

Reviewer 4



This paper discusses stein variational gradient descent and presents several theoretical results of it. The major contributions of this paper are: 1) constructing weak convergence of SVGD in the number of iterations and in the number of particles; 2) demonstrating a Vlasov continuous time limit; 3) providing an interpretation as a gradient flow of the KL divergence with respect to a new optimal transport metric. The techniques used in this paper are quite interesting and novel. These results help further understand the theoretical and intrinsic properties of SVGD. The paper is very well written, though there are a few grammatical errors. Some references should be updated. Overall, it is a good paper for NIPS publication. My only concern is that it is not clear whether we can get practical value out of these results. Will these results shed light shed light on new algorithms or improvements of existing ones? I hope the authors provide some further discussion of it.